# Embodied Carbon Premium for Cantilevers

**James Helal [1,*], Dario Trabucco [2], David Ruggiero [3], Paola Miglietta [4] and Giovanni Perrucci [2]**

1 Faculty of Architecture, Building and Planning, The University of Melbourne, Melbourne, VIC 3010, Australia
2 Department of Architecture and Arts & CTBUH Research Office, IUAV University of Venice, 30135 Venice, Italy; trabucco@iuav.it (D.T.); gperrucci@iuav.it (G.P.)
3 Concrete Behaviour and Structural Design Laboratory, École Polytechnique Fédérale de Lausanne, 1015 Lausanne, Switzerland; david.ruggiero@epfl.ch
4 Resilient Steel Structures Laboratory, École Polytechnique Fédérale de Lausanne, 1015 Lausanne, Switzerland; paola.miglietta@epfl.ch
* Correspondence: james.helal@unimelb.edu.au

**Abstract:** This study addresses the critical need for sustainable architectural designs within the context of climate change and the significant role the built environment plays in greenhouse gas emissions. The focus of this paper is on understanding the influence of unbalanced cantilevers on the embodied carbon of structural systems in buildings, a subject that has, until now, remained underexplored despite its importance in architectural innovation and environmental sustainability. Employing a case study approach, the Melbourne School of Design (MSD) building serves as a primary example to assess the embodied carbon implications of cantilevered versus supported structures. The methodological framework encompasses a comparative embodied carbon assessment utilising an input–output-based hybrid life cycle inventory analysis approach. The findings reveal that unbalanced cantilevers in buildings, exemplified by the MSD building, can lead to a 10% increase in embodied carbon compared to alternative designs incorporating supporting columns. Such findings underscore the environmental premium for cantilevers, prompting a re-evaluation of design practices towards minimising embodied carbon. Through this investigation, the research contributes to the broader discourse on sustainable construction practices, offering valuable insights for both design practitioners and educators in the pursuit of improving the environmental performance of the built environment.

**Keywords:** embodied carbon; sustainability; architectural design; cantilever; structural systems; buildings

## 1. Introduction

The need to address climate change is becoming increasingly urgent as human activities continue to intensify its effects. The Intergovernmental Panel on Climate Change (IPCC) report underlines that human-induced greenhouse gas (GHG) emissions, fuelled by population and economic growth, have surged to their highest levels in history [1]. Notably, the built environment plays a pivotal role in this context, accounting for approximately 40% of total global energy use and GHG emissions [2]. This sector's impact is further amplified by ongoing trends in urbanisation and population growth, which will likely escalate environmental impacts. Consequently, the building sector emerges not only as a significant contributor to climate change but also as an essential area for the implementation of rapid and substantial GHG emission reductions, often leveraging existing technologies and practices [1].

Historically, efforts to enhance the environmental performance of buildings have predominantly concentrated on operational carbon, which refers to the GHG emissions arising from the daily functions of buildings. Recently, however, a growing body of research has underscored the significance of embodied carbon. This term refers to the emissions linked to the extraction of raw materials, the manufacturing of building components and their transportation, construction activities, and the ongoing maintenance and replacement of

these materials throughout the building's life span. Indeed, in certain scenarios, embodied carbon can represent up to 60% of a building's total life cycle carbon footprint [3,4]. Consequently, a thorough understanding of both operational and embodied carbon becomes imperative for comprehensive environmental stewardship within the building sector. Moreover, this recognition necessitates that design decision-making processes in architecture be acutely cognisant of the embodied carbon implications, ensuring that sustainability is integrally considered from the inception of a project.

In this context, structural systems for buildings are identified as significant contributors to embodied carbon, accounting for an estimated 24% of a building's total life cycle embodied carbon emissions, according to benchmarking research [5]. The materials and processes involved in creating these systems, especially in innovative architectural designs, often entail significant carbon emissions [6]. This acknowledgment necessitates a paradigm shift in structural design frameworks, focusing not only on structural integrity but also on reducing embodied carbon. The quest for sustainable structural solutions demands innovative approaches that reconcile architectural vision with environmental responsibility, ensuring that the pursuit of aesthetically pleasing and functional structures does not come at the expense of environmental sustainability.

Recent research has elucidated the notion of "embodied carbon premiums" linked to various architectural features, such as height [7–16], slenderness [17], irregularity [18], and wide-open spaces [17,19]. For example, the construction of taller buildings typically requires an increased quantity of structural materials to counteract the intensified wind and seismic forces, a phenomenon referred to as the 'premium for height' [20]. Additionally, architectural designs that feature wide-open spaces tend to reduce the structural efficiency of beams and slabs, invariably leading to a considerable increase in material usage and, consequently, in embodied carbon [17]. Despite the growing body of knowledge in these areas, the embodied carbon premium associated with cantilevers, a common architectural element used for creating protruding overhanging structures, has not been thoroughly investigated until now. Such research is imperative for a deeper understanding of the environmental impacts prevalent in contemporary architectural practices and the effective mitigation of these impacts.

### 1.1. Aim and Scope

The primary aim of this paper is to explore the embodied carbon implications of unbalanced cantilever structures within buildings, employing a case study methodology centred on the Melbourne School of Design (MSD) building. By examining this specific instance, the research seeks to provide insights into how cantilevered versus supported structures can influence the embodied carbon of structural systems in buildings.

This study is grounded in the critical understanding of the relationship between GHG emissions and global warming potential (GWP). It adopts a comparative embodied carbon assessment approach, which simplifies the complexities inherent in a comprehensive life cycle assessment (LCA) while maintaining the accuracy necessary for informed decision making regarding resource utilisation and environmental impacts [21,22].

The methodology of this study is anchored in the adoption of standardised building life cycle stages as defined by the EN 15978 standard [23], which provides a framework for quantifying the environmental impact of buildings throughout their different life cycle stages. The stages are categorised into the product stage (A1–A3), which includes raw material extraction, processing, and manufacturing; the construction stage (A4–A5), which involves transport to the site and installation; the use stage (B1–B7), which encompasses use, maintenance, repair, replacement, and refurbishment; and the end-of-life stage (C1–C4), which covers deconstruction, waste processing, and disposal. The delineation of these stages allows for a structured and comprehensive assessment of the embodied carbon emissions associated with the different phases. Figure 1 graphically represents these stages, clearly delineating and emphasising the specific stages that are the focus of this study. The

subsequent discussion provides a thorough rationale for their inclusion and pertinence to the overarching objectives of this research.

| Product stage (A1-A3) | Construction stage (A4-A5) | Use stage (B1-B7) | | End of life stage (C1-C4) | Benefits and loads beyond the system boundary (D) |
|---|---|---|---|---|---|
| Raw material supply A1 | Transport A4 | Use B1 | Operational energy use B6 | De-construction demolition C1 | Reuse-recovery-recycling potential D |
| Transport A2 | Construction installation process A5 | Maintenance B2 | Operational water use B7 | Transport C2 | |
| Manufacturing A3 | | Repair B3 | | Waste processing C3 | |
| | | Replacement B4 | | Disposal C4 | |
| | | Refurbishment B5 | | | |

**Figure 1.** Scope of this study according to EN 15978 [23].

In structural engineering, the design emphasis is traditionally on achieving the functional objectives of structural systems with minimal recurrent maintenance and repair requirements. This approach inherently implies that the recurring embodied carbon, occurring during the operational phases (B1–B5: use, maintenance, repair, replacement, and refurbishment) of structural systems, is negligible. This is attributed to the durability and low-maintenance requirements typically inherent in well-engineered structural components and systems. Further, research indicates that the construction stages, A4 (Transport to the building site) and A5 (Installation into the building), generally account for only 1 to 3 percent of the total life cycle emissions [5,8], underscoring their minimal contribution.

Moreover, existing studies suggest that the end-of-life stages (C1–C4: de-construction; transport to waste processing; waste processing for reuse, recovery, and/or recycling; disposal) of structural systems contribute minimally to their total embodied carbon [24–27]. This minimal impact is largely due to the efficiency of contemporary deconstruction practices. Therefore, while these stages are an integral part of a building's overall life cycle, their influence on embodied carbon is relatively minor compared to the initial stages. Consequently, the embodied carbon assessment in this paper focuses predominantly on the product stage (A1–A3), acknowledging that the operational, construction, and end-of-life stages, although integral, have a comparatively smaller impact in terms of embodied carbon for structural systems.

*1.2. Notions and Definitions*

A structural element is a distinct and identifiable part of a structure. There are various types of structural elements, including columns, beams, slabs, and walls. These elements are fundamental components that contribute to a building's stability and integrity. The strategic arrangement of these structural elements form what is known as a structural system. This system is designed to effectively bear and distribute loads, ensuring the strength and stability of the structure. The structural system's design is crucial as it determines how the building will withstand various forces, including gravitational and lateral loads.

A unique form of a structural system is the cantilever, which is designed to extend outward from a single point of support, creating an overhanging structure. Unlike typical

beams, which are supported at both ends, a cantilever is only supported at one end. This distinctive design allows for the creation of extended spaces without the need for additional support, offering a degree of architectural flexibility and aesthetic appeal.

To achieve stability and balance, cantilevers require specific structural considerations. They necessitate rotational fixity or continuity at the supported end, along with vertical restraint, to ensure stability and load transmission. Contrary to initial impressions, a cantilever with sufficient boundary conditions can be structurally comparable to half of an inverted beam. The natural balance observed in tree branches extending from a trunk serves as an organic analogy for this principle.

In the context of building design, unbalanced cantilevers present more complex structural challenges. They exert increased forces on the building's core, necessitating enhanced structural support. Additionally, deflection at the cantilever's tip, often a primary design consideration, is amplified by the accumulated deformations from the back-span structure. The resulting overturning moments must be effectively transferred to the ground, often leading to more intricate foundation systems. The advent of modern structural materials such as steel and concrete has inspired architects to incorporate increasingly unbalanced, long-span cantilevers into their designs. While structural engineers are typically able to meet the demands of building codes for such structures, the additional structural requirements generally result in the use of greater quantities of materials. Consequently, this could lead to an increase in the embodied carbon of the structural system.

This paper specifically examines the embodied carbon premium associated with large, unbalanced cantilevers. While the focus is on a particular structure, the findings provide insight into the embodied carbon implications resulting from such architectural decisions. The investigation also explores alternative structural approaches that align more closely with natural load paths, promoting a more sustainable integration of structure-informed architectural design.

### 1.3. Structure of the Paper

This paper is structured into eight distinct sections. Section 1 articulates the research objectives and delineates the scope of the study, followed by a detailed exposition of key concepts and terminologies relevant to cantilever structures and their resulting environmental implications. Section 2 conducts an extensive review of the existing literature, focusing on the influence of cantilevers on material efficiency. This section also examines the prevalence of cantilevers as a significant architectural trend, drawing upon various illustrative examples. In Section 3, the paper presents a case study of the MSD building. This section elucidates the architectural rationale underpinning the use of cantilevers in the building's design and offers a detailed description of its structural system. Section 4 presents the study's research methodology, detailing the formulation of an alternative design scenario that incorporates supporting columns, alongside the methodology utilised to quantify the embodied carbon of the structural systems in both cantilevered and supported designs. The findings of the research are detailed in Section 5, which presents a comparative analysis of the embodied carbon across the design scenarios. Following this, Section 6 engages in a critical discussion of these results, contextualising them within the broader framework of architectural design and sustainability while also acknowledging the limitations inherent in the study. Section 7 delves into the broader ramifications of the research outcomes, elucidating their pertinence and consequentiality for key stakeholders in the fields of architecture, engineering, and design pedagogy. The paper culminates in Section 8, which offers a synthesis of the principal conclusions drawn from the research and suggests directions for future investigations into the embodied carbon implications of cantilever structures in the realm of building design.

## 2. Cantilevers in Buildings

### 2.1. Literature Review

Studies on cantilever behaviour can be traced back to Leonardo da Vinci and Galileo Galilei [28]. Despite this extensive history and the prevalent use of cantilevers in modern architecture, research focused on the material efficiency and embodied carbon implications of cantilevered structures remains scarce [29]. However, recommendations for minimising embodied carbon in structural systems commonly highlight the importance of optimising structural form and reducing material use early in the design process [30,31].

Several studies suggest that adopting simpler structural systems and shorter spans can lead to a more material-efficient design, yet they do not quantify the impact of structural system complexity on this efficiency [32–35]. Existing research has explored the embodied carbon associated with long-span structures, including balanced cantilevers, but has not explored the implications of unbalanced cantilevers or compared these with symmetric or supported structures [19]. The literature has a gap, with there being no systematic investigations available on the specific effects of unbalanced cantilever designs on the embodied carbon of structural systems for buildings.

Several studies have examined cantilevers, primarily focusing on engineering feats and architectural innovation, without critically examining the use of structural materials and their associated embodied carbon [36–40]. This trend in architectural and structural engineering discourse seems to prioritise design innovation, potentially overlooking considerations of environmental sustainability. Consequently, this paper seeks to contribute to the dialogue by exploring the embodied carbon implications of unbalanced cantilevers versus supported structures. The aim is to promote a more balanced approach to architectural practices, advocating for the integration of environmental considerations alongside architectural and structural innovation.

### 2.2. Cantilevers as an Architectural Trend

Cantilevers have become a significant element in the evolution of architectural design, introducing a dynamic aspect to the volumetric composition of buildings. Historically, the use of cantilevers was limited by the tensile and compressive strengths of traditional building materials such as stone, bricks, concrete, timber, and adobe. Timber, with its favourable tensile properties, was predominantly used for cantilevers in historical structures, facilitating the notable overhangs seen in the cantilevered volumes of Chinese dougong temples and medieval European timber-framed buildings. The introduction of steel in the 19th century transformed cantilever construction, allowing for significantly longer spans, as exemplified by cantilever bridges with spans of up to 100 m. This evolution reflects a shift from the necessity of material properties to the pursuit of architectural innovation and expression.

The adoption of reinforced concrete and steel in modern construction has made cantilevered elements a common feature, often being employed not just for their functional benefits in overcoming site constraints but also for their aesthetic appeal. This trend is highlighted in Figure 2, which showcases such examples: the Lamar Construction Headquarters, Stefano Boeri's Villa Méditerranée, and Richard Rogers' Drawing Gallery. Despite the absence of official databases tracking cantilever dimensions in buildings, Guinness World Records such as the "longest cantilever roof" of the Busan Cinema Center (South Korea) [41] and the "tallest cantilevered building" of the Central Park Tower (United States) [42] signify a competitive spirit in architectural design, pushing the boundaries of engineering and design innovation.

As seen in Figure 2a, the Lamar Construction Headquarters, designed by Integrated Architecture, showcases a striking 34.3 m cantilever that constitutes 90% of the building's length. This design choice, leveraging a heavily reinforced concrete core and a 23.3 m long steel truss, emphasises the client's construction expertise by opting for an architectural form that could have been simplified with load-bearing columns. The resultant structure demonstrates a deliberate departure from conventional and material-efficient designs to

achieve a visually compelling and structurally bold statement. The cantilever's rigidity negated the need for the initially planned tuned mass dampers in the final construction, underlying the potential over-engineering of the original design [37].

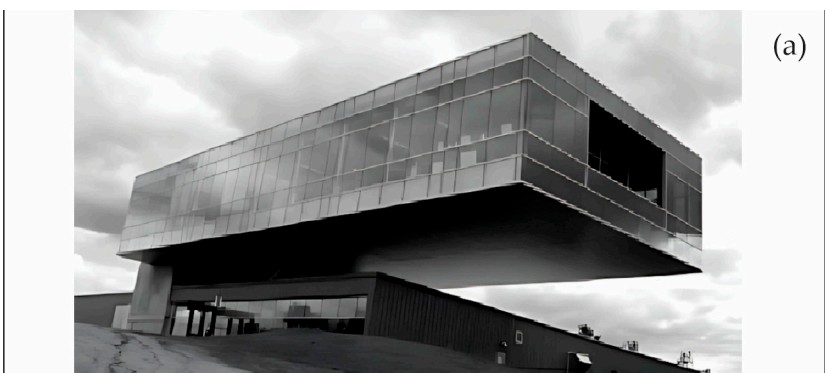

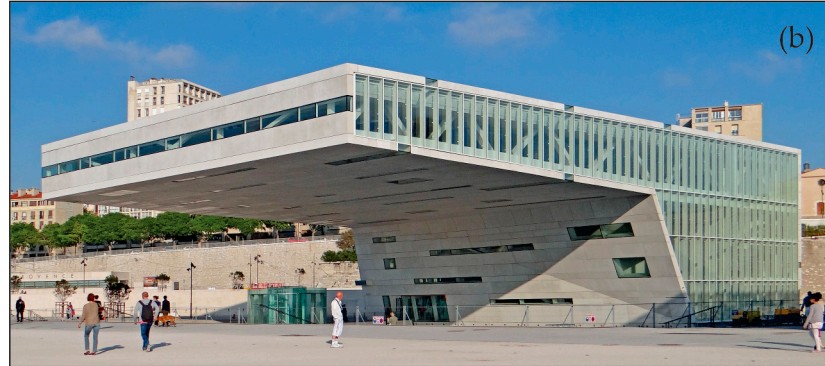

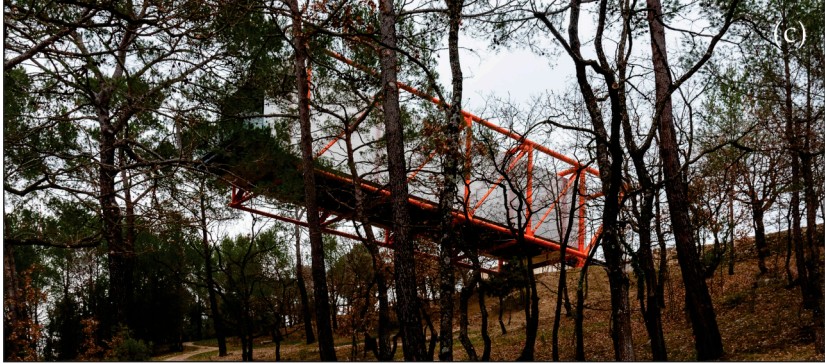

**Figure 2.** (**a**) Lamar Construction Headquarters, Michigan, United States. Reprinted with permission from Setareh et al. [43]; (**b**) Villa Méditerranée, Marseille, France. Reprinted with permission from Dalbéra [44]; (**c**) Chateau La Coste Drawing Gallery, Le Puy-Sainte-Réparade, France. Reprinted with permission from Poggi [45].

Stefano Boeri's Villa Méditerranée (see Figure 2b) introduces an appealing interaction with the aquatic environment through a 36-metre cantilever that covers 65% of the building length. The design incorporates a layer of sea water between two horizontal building elements, connecting an underwater conference centre with a cantilevered exhibition room. Notably, the entire area of the cantilevered exhibition hall is positioned directly above the underwater conference space, suggesting that additional supporting elements could have been strategically placed to diminish the need for such a pronounced structural system.

Shown in Figure 2c, the Chateau La Coste vineyard is home to a 120 square-metre art gallery, notable for its extension over a valley through a cantilevered steel frame extending from a concrete podium. Designed by Richard Rogers, an esteemed figure in architecture and 2007 Pritzker Prize laureate, the Richard Rogers Drawing Gallery is distinctively

supported by a bright orange steel frame, tethered to the cliff's edge by a steel hinge and tension cables that anchor it to the concrete foundations. The necessity for a substantial steel frame and extensive foundations raises questions about the environmental and material efficiency of placing a relatively modest-sized pavilion in such a complex position. While the site's constraints may have influenced the design choice, opting for a cantilevered structure over a traditionally supported one does not alter the internal experience or visibility from below, given the inaccessibility and natural coverage of the site. This case highlights the critical balance between aesthetic considerations and the practical implications of material use and site adaptability.

These examples, while showcasing the architectural and structural ingenuity inherent in cantilevered designs, also highlight the necessity for a more critical examination of their environmental implications. As architectural and engineering discourse continues to evolve, the integration of sustainability considerations into the design and implementation of cantilevered structures will be imperative.

## 3. Melbourne School of Design Building: A Case Study

The University of Melbourne, a distinguished institution in Australia, has played a pivotal role in shaping architectural education and practice through its Faculty of Architecture, Building, and Planning. This faculty houses the Melbourne School of Design (MSD), with the building itself, named after the school, constructed in 2017. The conception of the MSD building was a significant undertaking, aligning with the University of Melbourne's broader strategic vision. The 2005 'Growing Esteem' strategic plan set the foundation for this initiative, proposing changes in pedagogic strategy, graduate school establishment, and enhancing the on-campus student experience [46].

### 3.1. Design Intention

The process of selecting architects for the MSD building involved an international competition, reflecting the project's significance and its potential to set new standards for campus facilities. The competition's evaluation criteria focused on "built pedagogy," reflecting the project's aim to provide a learning environment that showcases the best of each profession represented by the faculty [47]. As seen in Figure 3, the building was envisioned as a blend of outstanding architecture, urban design, advanced construction techniques, and integrated design between the natural and built landscapes.

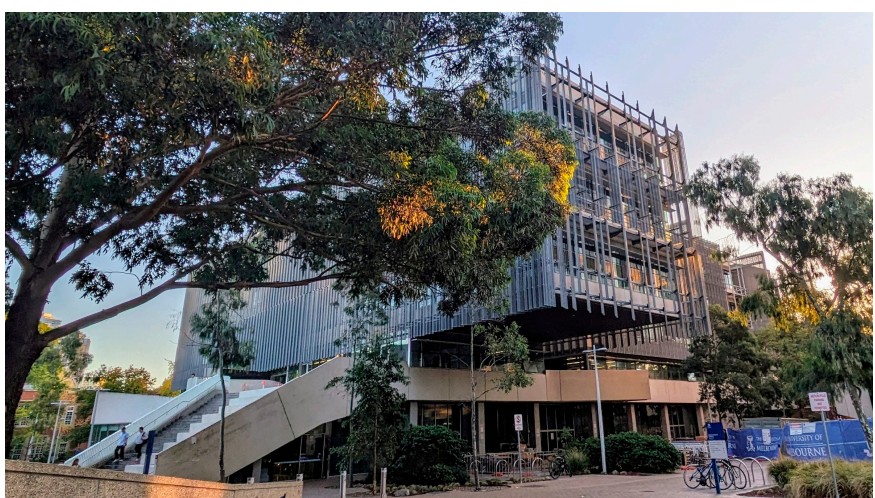

**Figure 3.** Melbourne School of Design (MSD) building, Melbourne, Australia.

The term "built pedagogy" is extensively adopted in the academic literature, informing the contemporary planning and design of educational spaces and places. Built pedagogy is defined as the "architectural embodiments of educational philosophies" [48]. This concept

implies that the physical structure of a building and its internal spaces not only reflect but also actively influence the educational philosophy and methodologies. These environments are not merely passive settings but dynamic participants in the teaching and learning process [49–51]. The Sir Owen G Glenn Building (OGGB) of the University of Auckland Business School [52] is another example where built pedagogy has been articulated and deliberately integrated into its design criteria, similar to the MSD building.

A study exploring the perspectives of the users and design team on the MSD building identified key themes in built pedagogy relating to institutional legibility and pedagogical practices [53]. The design team highlighted the evolving role of universities in a global context, particularly with the rise of online education and its impact on campus planning and facility design. The intention was for the MSD building to become a distinctive "destination for education." However, the pursuit of architecturally spectacular structures under the banner of built pedagogy, especially in architectural schools, has raised critical contradictions [54]. While aiming to enhance institutional legibility and global visibility, such buildings often feature distinctive architectural styles and features, as seen in RMIT's Swanston Academic Building and Design Hub, the Cube at the Queensland University of Technology, Ravensbourne University in London, IT University of Copenhagen, and AUB's Issam Fares Institute for Public Policy & International Affairs. The drive for such 'trophy buildings' can paradoxically run counter to the principles of sustainable architecture. These buildings, despite their visually captivating and architecturally distinctive styles, could possess a significant embodied carbon footprint, attributable to their bold yet often inefficient structural systems.

The unfolding climate emergency necessitates a re-evaluation of these architectural ambitions, especially in the context of architectural education. The challenge lies in balancing the desire for iconic buildings with the urgent need to minimise embodied carbon in response to climate change. This dichotomy is strikingly evident in the case of the MSD building, which, despite its pedagogical intentions, has the highest embodied carbon per square metre among the University of Melbourne's buildings that have undergone a life cycle assessment [55].

Such contradictions present an opportunity within the framework of built pedagogy: rather than serving purely as exemplars of architectural innovation, these structures can also be utilised as counterexamples. They can be integral to teaching future architects, engineers, and planners about the environmental impacts of building design choices and the importance of prioritising sustainability alongside aesthetic and functional considerations. This approach could foster a more holistic understanding of architecture, where environmental stewardship becomes as valued as architectural innovation.

### 3.2. Description of the MSD Building's Structural System

The structural system of the MSD building, shown in Figure 4, primarily consists of an exposed reinforced concrete frame. This system includes precast columns and core walls, which support in situ post-tensioned beams and reinforced concrete slabs.

As can be seen in Figure 4, a prominent feature of the MSD building is its unbalanced cantilever structure, which forms the main aspect of this study. Comprising three floors, this structure houses staff offices that project 12 m over the building's northeastern courtyard, representing 20% of the building's length. Constructed with a steel frame, the cantilever employs diagonal hangers on each flank, which redistribute the loads to the building's main reinforced concrete frame. Hanger tension loads are transferred across the building at the fifth floor, utilising post-tensioning tendons to connect to a core, highlighted in yellow in Figure 4, located on the opposite side of the building.

Regarding materiality, the building's foundation and sub-structure incorporate concrete with 50 MPa strength for bored piers and piles and concrete with 32 MPa strength for other elements like pile caps, pad footings, strip footings, ground beams, and slabs on the ground. The superstructure utilises 32 MPa concrete for interior slabs and beams, 40 MPa

concrete for exterior slabs, and higher-strength concrete (up to 65 MPa) for columns in the basement and ground levels, reducing to 50 MPa from level 1 to 4.

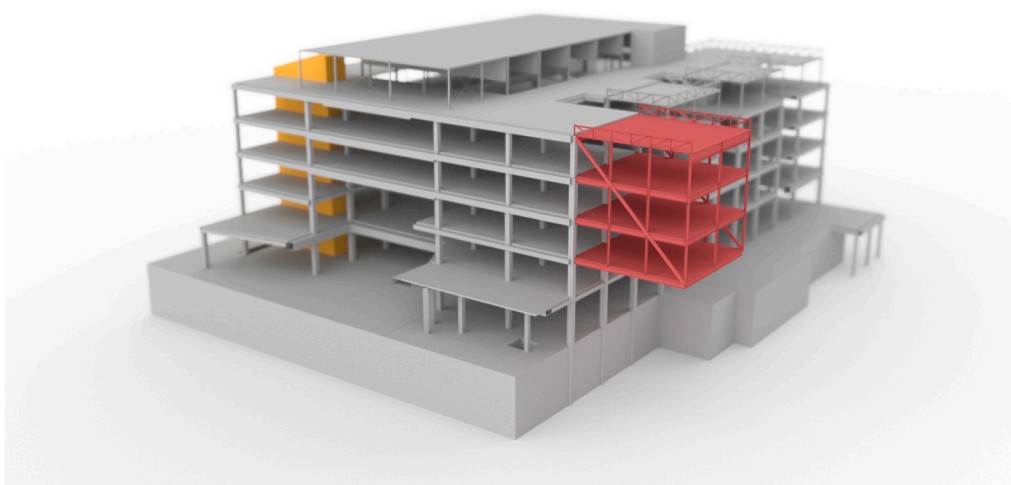

**Figure 4.** Three-dimensional render of the structural system of the as-built MSD building, highlighting the unbalanced cantilever in red and the core to which it is connected by post-tensioning tendons in yellow.

Regarding the structural design considerations, the MSD building adheres to the Australian structural design standards. It is designed to withstand wind loads, characterised by ultimate wind speeds of 46 m per second and serviceability wind speeds of 37 m per second. Additionally, the building is designed to withstand earthquake loads, factoring in an importance level of 3, a hazard factor of 0.08, and a site subsoil classification of B. In terms of fire safety, the building's structural components are designed to maintain integrity for a fire resistance period of 90 min.

## 4. Research Method

Through structural design, modelling, and analysis, design iterations can be explored, elucidating the influence of cantilevers on the embodied carbon of structural systems for buildings.

### 4.1. Design Scenarios

This study employed two distinct building models to examine the influence of cantilevers. The first model, referred to as the Base Case Model, accurately replicates the existing structure of the MSD building, complete with its notable unbalanced cantilevered structure. This model served as the benchmark against which the design alternative was compared, providing a baseline for understanding the embodied carbon of the building as it currently stands.

In the second model, known as the Supported Cantilever Model, the design was altered by introducing two supporting columns beneath the previously unbalanced cantilevered structure. As can be seen in Figure 5, this modification transforms the cantilever into a supported structure while keeping the rest of the building's design constant. This design alternative allows for an examination of how structural changes to the cantilever affect the building's embodied carbon.

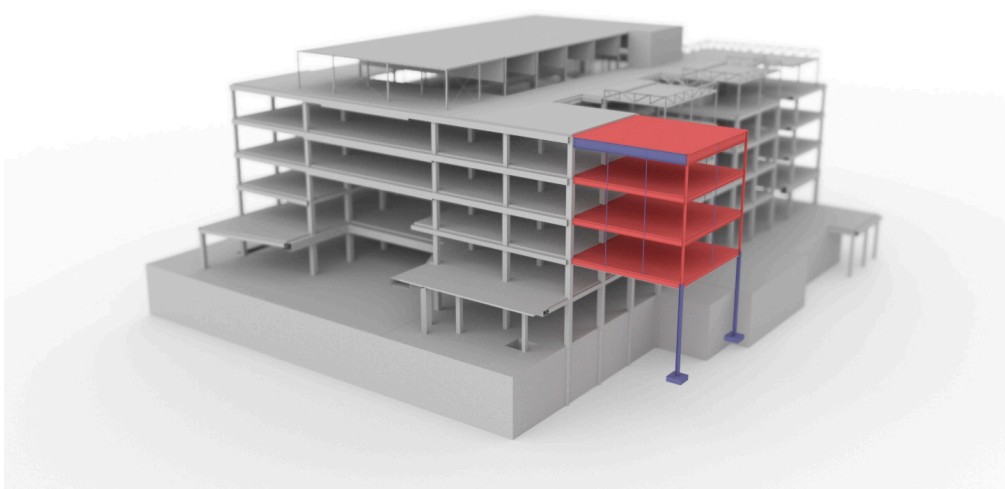

**Figure 5.** Three-dimensional render of the structural system of the alternative MSD building, highlighting the supported cantilever in red with its added structural members in blue.

### 4.2. Structural Modelling

For the quantification of the cantilever's effect on the structural system, a model of the as-built MSD building was developed using the commercial finite element software ETABS v21.2.0 [56]. The model aimed to replicate the actual structure as closely as possible, utilising information from structural drawings to balance trueness and analytical simplicity. Certain elements that have no bearing on the structural behaviour relevant to the influence of the cantilever were omitted for clarity. These include elements like hanging studios, bridges, and stairs, which do not contribute to the load paths of the cantilever being analysed.

The model incorporated steel and reinforced concrete column and beam elements (including post-tensioned beams) as line entities, reflecting the specifications found in the structural drawings. Similarly, shell elements were used to represent slabs, decks, and wall cores. The precast cladding of the MSD building, while not structurally integral, was accounted for by incorporating its weight as a line load along the edges of the corresponding slabs. This load was determined based on available precast cladding data, factoring in an assumed void area of 50%, which equates to a load intensity of 2.5 kN/m$^2$. The perforated sunscreen present on the façade was also accounted for as a line load corresponding to a weight of 0.5 kN/m$^2$ over its area based upon the manufacturer's information. Superimposed dead and live loads were applied in accordance with the structural drawings, ensuring that the model reflected the building's design loads accurately.

Wind and seismic forces were incorporated in accordance with the local standards outlined in AS1170.2 [57] and AS1170.4 [58], utilising parameters extracted from the structural drawings in concert with the Australian Standards. By correlating these values with the stipulations of AS1170.0, the structure's design life was inferred to be 50 years. To simulate the seismic response of the structure, a linear dynamic analysis using a response spectrum method was employed. Additionally, the section properties of the concrete elements were modified to account for anticipated cracking, as prescribed by the Australian Standard AS3600 [59], based on the verification of their stress levels under the most unfavourable non-seismic combination.

The development of this structural model established a baseline for the analytical assessment of forces acting upon various elements and provided a reference point for the alternative design. For the alternative building model described in Section 4.1, modifications were made judiciously, with steel columns being introduced to support the cantilever, leading to a redistribution of loads within the structure.

Introducing additional columns primarily served to mitigate the significant bending moment transferred to the building's lateral load resisting system. This alteration in structural behaviour facilitated the removal of one core wall and its extensive foundation

raft from the structural framework. However, it is imperative to emphasise that the core's role as an egress path must be preserved; thus, the elimination of the core's concrete material necessitates the incorporation of a stair enclosure that complies with building codes.

The modifications necessitated additional structural elements to provide support previously offered by the removed core. These changes included the addition of beams and a new column, accompanied by its foundation. As seen in Figure 6b, stability was maintained through structural adaptations of the cantilever (now supported and no longer a cantilever), such as replacing the parapet truss with a stiffer edge beam and removing the diagonal braces, supplemented by efficient hanger columns. In all cases, the structural steel elements were designed according to AS4100 [60].

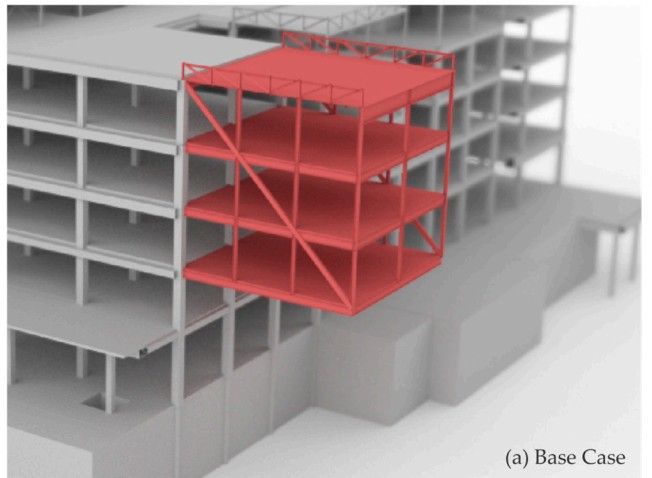 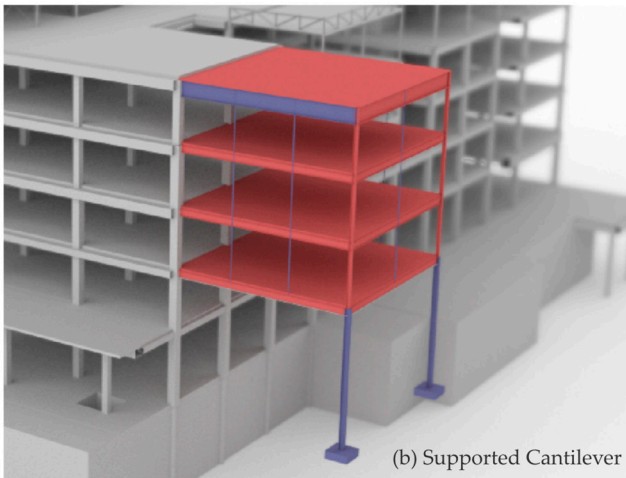

(a) Base Case  (b) Supported Cantilever

**Figure 6.** (**a**) Unbalanced cantilever of the Base Case Model of the MSD building; (**b**) supported cantilever of the alternative model of the MSD building, with the added structural members highlighted in blue.

The modelling process facilitated an examination of the changes in structural material volumes necessitated by both the as-built and alternative designs, enabling an approximate assessment that was carried out as follows:

- Adjustments were made to account for the net changes in the volumes of structural steel, ensuring an accurate reflection of modifications from the base case to the alternative design.
- Modifications in the volumes of reinforced concrete elements were calculated. This included accounting for the corresponding steel reinforcement and post-tensioning tendons, with adherence to specifications from structural drawings and employing average reinforcing rates for each structural element type.
- The elimination of the cantilever removed the necessity to counteract a significant tension force within the lateral system. Consequently, the flat post-tensioned tendons on Storey 5, previously compensating for this force, were excluded from the total material volume.
- Axial demand variations for all reinforced concrete columns were analysed between the two design options, using data from the structural models. Adjustments in the column sectional areas were made based on these demand changes, accurately reflecting increases or decreases in the total concrete volume.
- For the two remaining reinforced concrete cores, analyses of bending, shear, and torsion variations between the design options were conducted. This led to adjustments in the quantities of steel reinforcement required to meet the new demands. Bending moments were addressed by reconfiguring them into tension–compression couples in each direction, with the variation in force accounted for by the addition or subtraction of reinforcing steel. The shear design provisions of AS3600 [59] were employed to

calculate the necessary adjustments in transverse reinforcement for the shear and torsion demands.

These detailed considerations of the changes in structural material volumes between the base and alternative designs facilitated an informed calculation of the resultant change in embodied carbon.

### 4.3. Quantifying the Embodied Carbon of the MSD Building's Structural System

This section delineates the methodology employed in this study for calculating the embodied carbon of the structural system of the MSD building, including its base case and the alternative design. Given the significance of a comprehensive and consistent life cycle inventory analysis method in comparative embodied carbon assessments [12], this section elaborates on the methodologies chosen for our life cycle inventory (LCI) analysis.

There are three main approaches for compiling an LCI:

- The process analysis approach, which represents a bottom-up approach that studies the series of processes that, together, form the life cycle of a product or service. This detailed analysis, while offering high precision, might not encompass the entirety of complex supply chains, potentially leading to an underestimation of embodied carbon due to systemic incompleteness and various forms of truncation, including upstream, downstream, and sideways truncation [61];
- Environmentally extended input–output (EEIO) analysis offers a macro-economic perspective, correlating financial transactions with physical material flows [62,63]. It integrates data from economic input–output tables with environmental metrics, thereby enabling the quantification of environmental impacts throughout intricate supply chains. This method, while comprehensive, may lack the detail required to distinguish between different actors within an industry or products within a product group;
- Hybrid analysis, which merges process data with EEIO data to ensure both specificity and comprehensiveness in the analysis of products or services [64].

To mitigate the limitations of both process analysis and EEIO analysis, we firstly employed a hybrid LCI approach, specifically the Path Exchange (PXC) method, which has been shown to be the most efficient LCI method globally, maintaining comprehensive coverage of the system boundary [65–67].

To calculate embodied carbon, this study utilised the embodied carbon coefficients of the Environmental Performance in Construction (EPiC) database [68], compiled via the PXC method and particularly relevant to the Australian context. The embodied carbon of the MSD building's structural system, including the cantilevered sections, was calculated using the following equation:

$$EC_{SS} = \sum_{m=1}^{M}\left(Q_{m,SS} \times ECC_m\right) \tag{1}$$

where $EC_{SS}$ = embodied carbon of structural system SS in $kgCO_2$-e; $Q_{m,SS}$ = quantity of material m in structural system SS (e.g., steel in kg); and $ECC_m$ = embodied carbon coefficient of material m (e.g., 2.86 $kgCO_2$-e/kg for steel).

The EPiC database's hybrid LCI approach is particularly pertinent as it mirrors the economic and environmental context of Australia, thus ensuring the relevance and accuracy of the data. However, it is important to recognise that this approach assumes domestic production conditions for imported materials, a factor that might not fully represent the specific production conditions of materials from other countries. This assumption is a standard practice in LCI modelling and is considered a potential area for future refinement as global data and modelling capabilities evolve.

To evaluate the impact of LCI methodologies on the study's findings and conclusions, we carried out an additional analysis, recalculating the embodied carbon of the MSD building's structural system through the utilisation of process-based Environmental Product Declarations (EPDs). This approach entails compiling a comprehensive collection of EPDs from a variety of sources, including databases, program operators, and individual

suppliers. By deriving an average value for each material from the aggregated EPDs and then applying these values to the quantities ascertained from the structural modelling and analysis, this study can articulate both best-case and worst-case scenarios.

In this recalculated analysis, the process-based EPD coefficients were used in place of the EPiC database coefficients previously applied in Equation (1). This adjustment allowed for a nuanced comparison of embodied carbon outcomes based on different data sources, further enriching the study's insights into the embodied carbon implications of the MSD building's cantilevered section.

## 5. Results

This section details the results of our study, which focused on the influence of the unbalanced cantilever of the MSD building on the embodied carbon of its structural system. Material quantities from the analysed and designed structural models were gathered and converted into their embodied carbon values. Section 5.1 explores the embodied carbon of the Base Case Model of the MSD building's structural system. Section 5.2 examines the embodied carbon of the alternative design incorporating supporting columns. Furthermore, Section 5.3 presents insights from a sensitivity analysis evaluating the impact of life cycle inventory analysis on the results.

### 5.1. Embodied Carbon of the Base Case Model of the MSD Building's Structural System

Our analysis of the Base Case Model of the MSD building's structural system revealed a total embodied carbon of 7,197,606 $kgCO_2$-e. To provide a clearer perspective on the distribution of embodied carbon, this amounts to 689 $kgCO_2$-e/$m^2$ of net floor area. As detailed in Section 4.2, this calculation is derived from a modelling approach that balances accuracy with analytical simplicity. Thus, it is important to note that certain elements, such as hanging studios, bridges, and stairs, which do not directly influence the cantilever's structural behaviour, were not modelled.

As seen in Table 1, the distribution of embodied carbon by storey highlights significant variances, with the substructure (foundations and basement) contributing the most at 27.7%, followed by the ground storey at 19.3%. This finding underscores the pivotal role that foundation design plays in the environmental impact of building construction. Despite this, the area of foundation design, particularly its implications for embodied carbon, remains relatively underexplored in architecture and structural engineering studies compared to the extensive body of work on the embodied carbon of superstructures [69,70].

**Table 1.** Embodied carbon per storey of Base Case Model of MSD building structural system.

| Storey | Embodied Carbon ($kgCO_2$-e) | % of Total |
|:---:|:---:|:---:|
| Substructure [1] | 1,996,349 | 27.7% |
| Ground storey | 1,385,976 | 19.3% |
| Storey 1 | 921,626 | 12.8% |
| Storey 2 | 667,851 | 9.3% |
| Storey 3 | 731,959 | 10.2% |
| Storey 4 | 818,803 | 11.4% |
| Storey 5 | 506,310 | 7.0% |
| Roof | 168,732 | 2.3% |

[1] Substructure includes the basement storey as well as the foundations.

The examination of embodied carbon by material offers crucial insights into the predominant contributors to the building's carbon footprint. As shown in Table 2, concrete can be identified as the primary material, constituting 54.3% of the total embodied carbon of the MSD building's structural system. In contrast, steel—encompassing structural steel, steel reinforcement, post-tensioned tendons, and corrugated steel sheets—accounts for approximately 1.5% in volume relative to concrete's 95%. Despite this volumetric disparity, the contribution of steel to the embodied carbon of the entire structural system is

approximately 44.8%. This significant figure is attributable to the notably higher embodied carbon intensity of steel compared to concrete, a difference that is accentuated by the typical steel production process in Australia, which significantly relies on the Blast Oxygen Furnace (BOF) method. Unlike the Electric Arc Furnace (EAF) process, which uses recycled materials and tends to be less carbon-intensive, the BOF process—prevalent in Australia due to its legacy as a major means of iron ore extraction—contributes to the higher embodied carbon values observed. This production characteristic likely influences the EPiC coefficients, suggesting that the embodied carbon figures for steel in this study are reflective of specific Australian practices. Consequently, in regions where the EAF process is more common, the contribution of steel to the overall embodied carbon of the structural system might represent a smaller proportion. Such findings highlight the potential for substantial reductions in embodied carbon through enhancements in the material manufacturing processes for steel, underscoring the importance of focusing on both material selection and manufacturing efficiencies in mitigating the environmental impact of construction.

**Table 2.** Embodied carbon per material of Base Case Model of MSD building structural system.

| Material | Embodied Carbon (kgCO$_2$-e) | % of Total |
|---|---|---|
| Concrete | 3,909,356 | 54.3% |
| Steel Reinforcement [1] | 1,839,523 | 25.6% |
| Corrugated Steel Sheet | 974,993 | 13.5% |
| Structural Steel | 407,334 | 5.7% |
| Laminated Veneer Lumber | 66,401 | 0.9% |

[1] Steel reinforcement includes post-tensioning tendons, which account for 1.1% of the total embodied carbon of the MSD building's structural system.

The embodied carbon associated with different structural member types in building construction provides a clear depiction of where the most significant environmental impacts occur. As indicated in Figure 7, reinforced concrete (RC) slabs and beams each represent approximately 28% of the structural system's total embodied carbon, followed by the steel roof deck at 19%. In contrast to tall buildings, where lateral loads such as wind and seismic forces govern design and material utilisation, low-rise buildings like the MSD building are less affected by such forces. Consequently, the structural system's embodied carbon is predominantly influenced by the structural materials required to resist gravity loads. The 'Other' category seen in Figure 7, which accounts for 4% of the embodied carbon, includes miscellaneous elements that do not fall into the primary structural member categories (e.g., steel columns and beams supporting the hanging studios). While, individually, these elements may contribute less to the total, collectively, they represent a non-negligible percentage, suggesting that a comprehensive approach to embodied carbon reduction should also consider these additional components.

The central focus of this study is to evaluate the embodied carbon implications of the unbalanced cantilever within the Melbourne School of Design (MSD) building. Thus, Table 3 delineates the embodied carbon associated with the cantilever in comparison to the rest of the structural system.

**Table 3.** Embodied carbon of the cantilever within the MSD building's structural system.

| Structural Subsystem | Embodied Carbon (kgCO$_2$-e) | % of Total |
|---|---|---|
| Cantilever | 191,995 | 2.7% |
| Rest of Structural System | 7,005,612 | 97.3% |

At a cursory glance, the seemingly inconsequential 2.7% contribution of the cantilever to the total embodied carbon may induce the presumption that the influence of the cantilever on the embodied carbon of the structural system of the MSD building is negligible. Such an interpretation, however, would be misleading. A more comprehensive under-

standing necessitates comparing the embodied carbon of the MSD building's structural system equipped with the unbalanced cantilever against the alternative design devoid of it. The true measure of a cantilever's influence is encapsulated not merely in the cantilever's intrinsic embodied carbon but, more critically, in the difference between the embodied carbon of the structural systems with and without the cantilever.

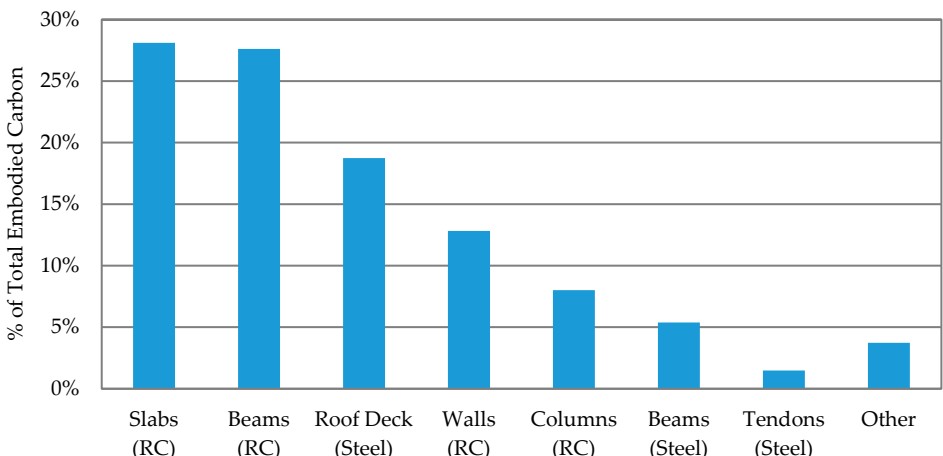

**Figure 7.** Embodied carbon per structural element type of Base Case Model of MSD building structural system.

### 5.2. Embodied Carbon of the Supported Cantilever Model of the MSD Building's Structural System

This section presents an analysis of the alternative design for the MSD building's structural system, which incorporates supporting columns, and contrasts this with the embodied carbon of the base case scenario.

The analysis herein focuses on the portion of the MSD building structurally affected by the cantilever—the eastern section of the building, which constitutes 20.2% of the building's total net floor area. It is critical to confine the analysis to this segment, as structural modelling indicates that the remainder of the structure, predominantly influenced by gravity loads, as identified in Section 5.1, remains relatively unaffected by the cantilever due to its specific load paths. Thus, for a meaningful analysis, the proportion of total embodied carbon should reflect only the segment of the structural system that the cantilever influences, as shown in Figure 8.

The embodied carbon for the structural system of the alternative MSD building design with supporting columns amounts to 7,039,696 kgCO$_2$-e. This figure represents a 2.26% reduction compared to the base case. However, when the analysis is refined to the cantilever-influenced segment of the structural system, accounting for 20.1% of the net floor area, the cantilever's impact on embodied carbon is revealed to be a 10.9% increase for that section. This increase underscores the substantial effect the cantilever has on the structural system's efficiency and its embodied carbon footprint.

As highlighted in Section 4.2, maintaining the building's egress pathways is essential; therefore, removing the concrete of the core necessitates adding a stair enclosure that adheres to building regulations and ensures structural integrity, particularly for a fire resistance duration of 90 min. To this end, a non-structural stair enclosure, constructed from concrete blocks measuring 390 × 190 × 140 mm with an embodied carbon of 2.7 kgCO$_2$-e per unit (according to the EPiC database), was incorporated into the Supported Cantilever Model. This addition led to an increase in embodied carbon of 12,807 kgCO$_2$-e, adjusting the overall reduction in embodied carbon attributable to the cantilever from 10.9% to 10%. Conducting this analysis is crucial to confirm that the savings in embodied carbon are genuinely realised across the entire building and not merely redistributed, thereby providing a more accurate assessment of cantilevers' impact on embodied carbon.

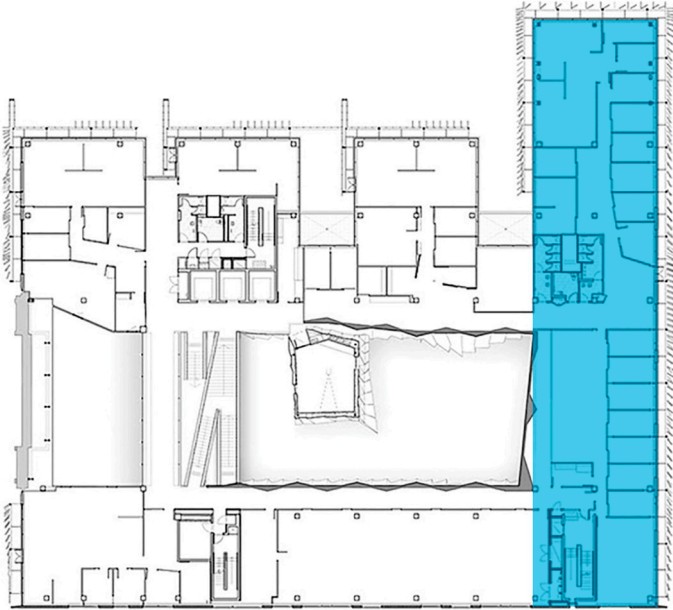

**Figure 8.** Floor plan of the MSD building's third storey, with the cantilever-influenced segment highlighted in blue.

Supporting the cantilever with columns led to a reduction in structural material volume by 121 cubic metres (comprising both concrete and steel). This measure not only signifies a tangible decrease in the number of materials transitioned into the technosphere but also underscores the wider environmental benefits spanning the entire life cycle—from the extraction of raw materials to the transportation of the manufactured construction materials and their eventual deployment on site. Highlighting the imperative of adopting comprehensive strategies to minimise environmental effects in the face of a climate crisis, this finding illustrates the significance of maintaining potential construction materials within the natural environment rather than their assimilation into the built environment.

The ensuing discussion will examine the factors contributing to this increase by investigating changes in embodied carbon per storey, per material, and per structural element type. This approach will elucidate the underlying reasons for the heightened embodied carbon attributable to the cantilevered design, thus integrating the findings into the broader context of the study and contributing to the discourse on sustainable structural design.

An evaluation of the foundation of the alternative building design showed a 4% reduction in embodied carbon across the entire foundation system, which increases to 19.9% when focusing solely on the cantilever-influenced structure. This indicates that apart from the core wall's removal, significant embodied carbon savings were achieved by eliminating the foundation requirements for the core wall. This interdependence between superstructure and substructure illustrates a critical point often overlooked in existing research: while the focus on the embodied carbon premium of various architectural decisions tends to concentrate on superstructures, it likely underestimates the full extent of these decisions on the overall embodied carbon premiums. This finding highlights the necessity for a more holistic approach in assessing the environmental impacts of architectural and structural design choices, ensuring that the influence on both superstructures and substructures is adequately considered.

Figure 9 further elucidates these findings, depicting the changes in embodied carbon per storey and highlighting the greater influence on the substructure compared to any of the storeys of the superstructure individually, thereby underlining the critical insights gained from our holistic evaluation of architectural and structural design impacts. An analysis of embodied carbon changes per storey of the cantilever-influenced segment of the structural system revealed that Storeys 2, 3, and 4 experienced a 10 to 12.5% change,

with other levels seeing approximately a 5% change. This differential impact across the storeys was expected given that the entire building is subject to the ramifications of the design change, yet the cantilever's physical presence on Storeys 2, 3, and 4 means these levels are more directly affected. The removal of structural elements previously necessary for the cantilever's support, due to the introduction of supporting columns, alters the material requirements of these specific storeys, underscoring their heightened influence on the overall embodied carbon variation.

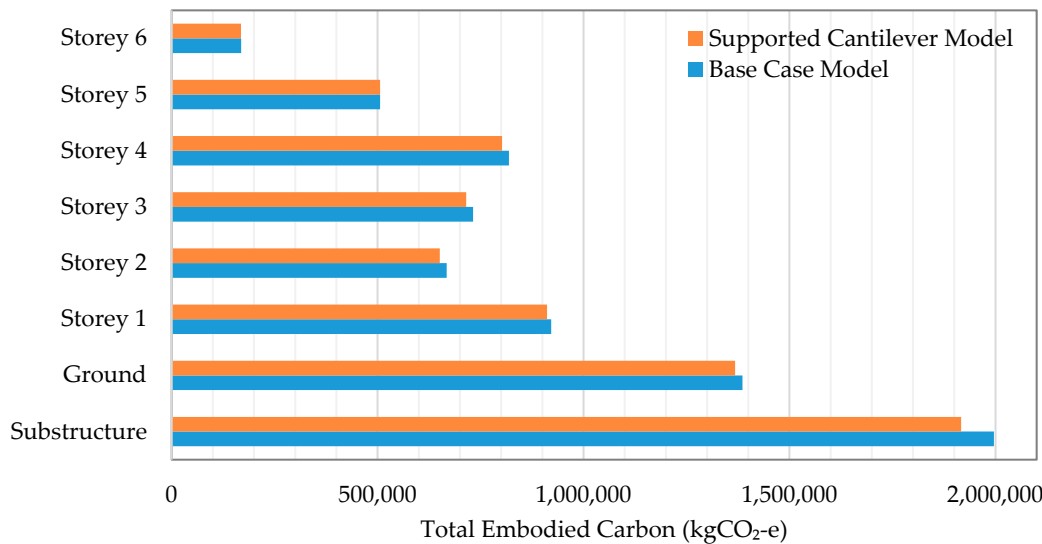

**Figure 9.** Embodied carbon per storey of the Supported Cantilever Model compared to the Base Case Model of the MSD building structural system.

As seen in Table 4, an examination of embodied carbon changes per structural element type indicated a notable 24% reduction in the embodied carbon of core walls due to the removal of one core associated with the cantilever in the as-built scenario. However, this adjustment led to an increase in the embodied carbon in the beams and columns, reflecting the redistribution of structural loads. This scenario underscores the inherent complexities within structural systems, where the removal of certain elements necessitates the introduction of additional structural supports elsewhere to accommodate for altered load paths. Such dynamics highlight the critical balance between removals and additions within a structure, emphasising the importance of evaluating whether these trade-offs result in a net reduction in embodied carbon. In this instance, the overall design adjustments did indeed culminate in a beneficial reduction, illustrating the potential for thoughtful structural modifications to enhance environmental outcomes.

**Table 4.** Percent change in embodied carbon per structural element type between the base case and the alternative design of the MSD building superstructure.

| Element Type | Embodied Carbon (kgCO$_2$-e) | %Δ |
| --- | --- | --- |
| Walls (RC) | 508,111 | −23.8% |
| Beams (Steel) | 1,475,485 | +5.2% |
| Beams (RC) | 974,993 | +2.7% |
| Columns (RC) | 424,453 | +1.8% |

Regarding materials, the decrease in the concrete's embodied carbon accounted for 46.6% of the total reduction, with steel (encompassing reinforcement and post-tensioning applications) comprising the remainder. This distribution highlights the critical role of all materials in achieving embodied carbon reductions, underscoring the comprehensive nature of sustainable design considerations. Figure 10 plots the embodied carbon per material for the Supported Cantilever Model versus the Base Case Model of the MSD

building's structural system, providing a visual representation of these reductions and their distribution among different materials.

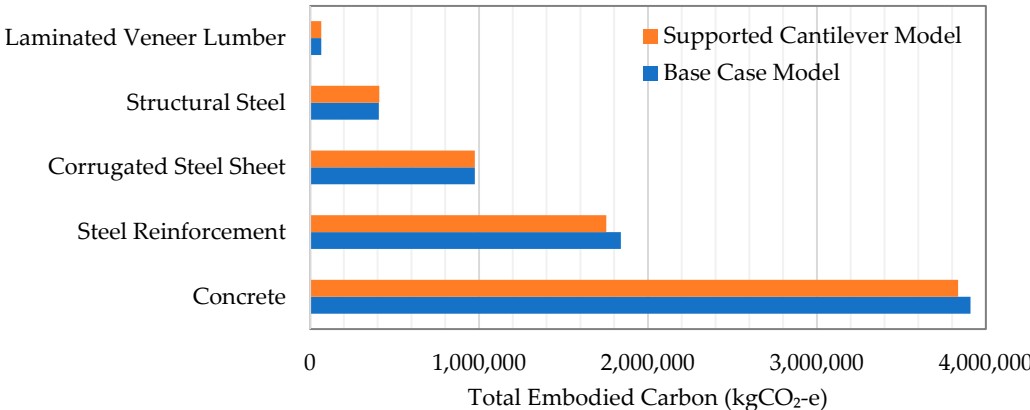

**Figure 10.** Embodied carbon per material of the Supported Cantilever Model compared to the Base Case Model of the MSD building structural system.

*5.3. Influence of Life Cycle Inventory Analysis Approaches on the Embodied Carbon Premium for Cantilevers*

Utilising the process-based EPD coefficients as a substitute for the EPiC database coefficients previously applied revealed significant insights. The recalculated total embodied carbon for the structural system of the Base Case Model of the MSD building was found to be 6,590,905 $kgCO_2$-e. Contrastingly, the use of embodied carbon coefficients from the EPiC database, which benefits from a comprehensive system boundary in its LCI approach, yielded an embodied carbon value of 7,197,606 $kgCO_2$-e. The embodied carbon per net floor area comparison revealed values of 631 $kgCO_2$-e/$m^2$ and 689 $kgCO_2$-e/$m^2$ when utilising average EPD values and EPiC database coefficients, respectively. Thus, the employment of average EPD values results in a calculation approximately 8.5% lower than when using EPiC database coefficients. This typical discrepancy underscores the critical need for consistency, transparency, and comprehensiveness in the LCI approaches underlying databases of embodied carbon coefficients for construction materials.

Applying the extreme values from the worst-case and best-case EPDs for the base case structural system results in an embodied carbon range that spans from 7,563,537 $kgCO_2$-e to 5,445,459 $kgCO_2$-e. This substantial variance among EPD sources translates to a deviation of more than plus or minus 15% from the calculated average. This variability underscores a prevalent issue within the realm of EPDs—their frequent failure to comply with mandatory information disclosure standards. Such non-adherence significantly hampers the comparability of these documents, which is crucial for accurate embodied carbon assessments. Previous research involving the scrutiny of hundreds of EPDs within similar categories found that 8.06% of the documents were entirely non-comparable, 89.15% were deemed incomparable, 2.75% could be compared with caution, and a mere 0.04% were directly comparable [71]. This lack of standardisation and transparency in EPD information not only complicates the evaluation of embodied carbon but also impedes the development of reliable and consistent sustainable design strategies.

When average EPD values are applied, the cantilever's embodied carbon constitutes 3.24% of the structural system's total embodied carbon, compared to the 2.67% identified using EPiC coefficients. The embodied carbon attributed to the substructure accounts for 25.7% of the total when utilising EPDs, a slight decrease from the 27.7% identified with EPiC coefficients. Moreover, the proportion of embodied carbon attributed to concrete shifts to 42.5% of the structural system's total when EPDs are employed, in contrast to the 54.31% derived from EPiC coefficients.

This variation in the proportion and dominance of components and materials in terms of embodied carbon highlights the significant impact of differing LCI methodologies on

the assessment of architectural decisions. The primary cause for this variation lies in the fact that embodied carbon coefficients do not only vary by material type but also follow a non-proportional and somewhat unexpected progression. For example, the EPiC database demonstrates a linear increase in embodied carbon coefficients for concrete with increasing strength grades, in contrast to the non-linear trend observed in aggregated EPDs. Here, coefficients rise until the 50 MPa grade and then exhibit a substantial decrease for the 65 MPa grade.

For the Supported Cantilever Model, utilising average EPD values results in the total embodied carbon being reduced to 6,451,792 $kgCO_2$-e, marking a 2.1% decrease for the entire structural system. Accordingly, the cantilever's influence on the embodied carbon of the segment affected remains consistent at approximately 10%, matching the findings obtained using EPiC coefficients. This consistency across different data sources for embodied carbon coefficients reaffirms the significant impact of cantilever designs on the embodied carbon of structural systems, highlighting the nuanced understanding required in assessing sustainable architectural practices.

## 6. Discussion

This investigation sought to examine the influence of cantilevers on the embodied carbon of structural systems, with the MSD building serving as a case study. Notably, the cantilevers observed in current architectural practice, as outlined in Section 2.2, generally extend longer both in absolute terms and relative to the building's length. Thus, the findings of this research, given its conservative approach, likely represent the lower bound of cantilevers' actual impact on embodied carbon.

The specific cantilever analysed contributed to 2.26% of the total embodied carbon within the structural system. However, its impact on the segment of the structural system affected by the cantilever, constituting approximately 20.1% of the building's floor area, was significantly more pronounced. In particular, the cantilever's influence on this segment was observed to exceed 10%, highlighting the substantial, and sometimes underestimated, effects of architectural features on a building's environmental footprint.

The disparities highlighted by the sensitivity analysis, particularly between the EPiC database and EPDs, underscore the complexities inherent in comparative embodied carbon assessments. These discrepancies call for the adoption of standardised practices and more rigorous life cycle assessment methodologies aimed at enhancing the consistency and reliability of embodied carbon evaluations. Moreover, the outcomes of this research illuminate the complex dynamics of embodied carbon assessments, offering both quantitative insights and fostering a deeper reflection on the intersections of architectural aesthetics, construction practices, and sustainability objectives.

Limitations inherent to this study include its focus on the specific structural materials, typology, and floor plan geometry of the MSD building, alongside the design strategies implemented in the alternative design proposal. Furthermore, the embodied carbon calculations are based on data reflective of Australia's economic conditions and energy profiles, suggesting that the findings have a geographical specificity. However, the structural material quantities derived from our structural analysis remain applicable to other areas that are exposure to similar lateral loads. Future research could adapt these quantities to different regional contexts by applying locally relevant embodied carbon coefficients, thus broadening the scope of the findings.

Prospects for future research extend beyond the current study's parameters, suggesting avenues for a more comprehensive exploration of the embodied carbon implications of cantilevers. Such investigations could include expanding the system boundary to encompass additional life cycle stages and integrating other building systems, thereby offering a more holistic view of embodied carbon considerations in architectural design.

## 7. Implications and Significance of the Research

This section delves into the broader implications of the research, both for design practitioners and pedagogues, underscoring the transformative potential of informed design choices and educational approaches in fostering a more sustainable built environment.

### 7.1. Implications for Design Practitioners

The urgent need to mitigate climate change through effective design strategies is more pressing than ever. The World Green Building Council (WGBC)'s forecast that embodied carbon will constitute 50% of the GHG footprint of new constructions between 2020 and 2050 underscores this reality [6]. As the production of steel and concrete, fundamental elements in building construction, contribute significantly to global GHG emissions, optimising structural designs emerges as a crucial strategy for meeting climate change mitigation goals.

This study's exploration of design scenarios, particularly focusing on unbalanced cantilevered structures, underscores the potential for more embodied carbon-efficient structural designs. The case of the MSD building, examined in this research, serves as a powerful illustration. While cantilevers may contribute to architectural aesthetics, their impact on embodied carbon is significant. These findings align with and support initiatives like the Structural Engineers (SE) 2050 Commitment Program [72], which urges design practitioners to adopt practices aimed towards reducing GHG emissions through design improvements.

The increasing legislative momentum towards reducing embodied carbon in construction projects further accentuates the importance of this study's findings. Globally, a growing number of countries are implementing or on the cusp of introducing legislative ceilings on the embodied carbon of new buildings [73–75]. These regulatory frameworks aim to ensure that construction projects contribute to national and international climate change mitigation efforts. In this context, the distinction highlighted by this study, such as achieving a 10% reduction in embodied carbon through specific design decisions like opting for supported structures over cantilevers, could become a pivotal factor in the approval process for building projects. Conversely, failing to consider the embodied carbon implications of design elements could jeopardise a project's viability by preventing it from meeting the required environmental standards. Thus, the findings of this study highlight the importance of considering embodied carbon impacts in the design phase, particularly with respect to architectural features such as cantilevers, which can significantly influence a project's compliance with emerging legislative requirements.

### 7.2. Implications for Design Pedagogues

In the realm of design education, this research brings to the fore the concept of "built pedagogy". The recent shifts in national standards of competency for architects by the Australian Institute of Architects, for example, incorporating a deeper understanding of embodied carbon, reflect a growing recognition of the importance of integrating embodied carbon considerations into design education and decision making [76].

Using existing buildings such as the MSD building as pedagogical tools is pivotal. Rather than exemplifying them as models of design excellence, they should be presented as both exemplars in some respects and counterexamples in others, demonstrating what practices should be avoided to reduce embodied carbon. This perspective aligns with the evolving concept of "built pedagogy", wherein the physical structure of a building serves as an educational tool not only for evaluating its design success but also for evaluating its environmental impact.

Incorporating these insights into design curricula could profoundly impact future architects and engineers. By critically analysing buildings like the MSD in educational settings, students could learn to appreciate the complex interplay between aesthetics, functionality, and environmental sustainability. This approach would encourage a shift in perspective, where sustainable design practices are not just an afterthought but a fundamental aspect of the design process.

## 8. Conclusions

The construction industry stands at a pivotal juncture, facing the imperative to adapt swiftly to meet urgent climate mitigation goals and address a broad spectrum of environmental challenges. Within this context, the significance of embodied carbon, often overlooked or undervalued in environmental assessments of buildings, emerges as a critical point of focus. Embodied carbon plays a crucial role in the early life cycle emissions of buildings, thus necessitating a strategic shift in architectural and engineering practices to prioritise its reduction from the design phase.

This paper focused on analysing the influence of cantilevers on the embodied carbon of building structural systems, using the MSD building as a case study. By employing a series of structural models and a hybrid life cycle inventory analysis, the study quantified the embodied carbon associated with cantilevered designs in comparison to supported structures. The investigation revealed that in the case of the MSD building, the unbalanced cantilever contributed to a 10% increase in the structural system's embodied carbon footprint. It is essential to note that this percentage could vary in other buildings, potentially exceeding 10% if cantilevers in those structures were longer or occupied a higher proportion of the building's total length. Thus, the study's findings, though potentially conservative, signal the broader implications of design choices on sustainability metrics, advocating for a more critical evaluation of architectural elements in the context of embodied carbon.

Amidst escalating urban development, the discourse on sustainable architectural and engineering practices is becoming increasingly relevant. The re-evaluation of cantilevers and similar architectural features presents a tangible opportunity to diminish upfront embodied carbon emissions, contributing to sustainable urban development objectives. This shift towards structurally efficient designs, leveraging existing knowledge and materials without necessitating new construction methodologies, underscores the feasibility of significant embodied carbon savings through thoughtful design.

Furthermore, this research resonates with the concept of "built pedagogy", particularly within architectural education, where the pursuit of visually striking "trophy buildings" often overshadows sustainability considerations. The findings underscore the importance of integrating sustainability into the pedagogical narrative, challenging architectural schools to transcend the allure of iconic buildings in favour of designs that embody environmental stewardship. By fostering an educational ethos that values sustainability as highly as aesthetic and functional innovation, future built environment professionals can be equipped to navigate the complexities of sustainable design, making informed decisions that favour the environment.

**Author Contributions:** Conceptualisation, J.H., D.T., and D.R.; methodology, J.H., D.T., D.R., and P.M.; software, D.R. and P.M.; formal analysis, J.H.; investigation, J.H., D.R., P.M., and G.P.; data curation, J.H.; writing—original draft preparation, J.H., D.T., D.R., and P.M.; writing—review and editing, J.H., D.T., D.R., P.M., and G.P; visualisation, J.H.; project administration, J.H.; funding acquisition, D.T. All authors have read and agreed to the published version of the manuscript.

**Funding:** The Article Processing Charges was funded by IUAV University of Venice.

**Data Availability Statement:** The finite element models developed for this study can be freely accessed on Figshare at https://figshare.com/s/c6f2f28682c7542633e2, accessed on 22 March 2024.

**Conflicts of Interest:** The authors declare no conflicts of interest.

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
