# Peer review of "Embodied Carbon Premium for Cantilevers"

_buildings, doi:10.3390/buildings14040871_

Round 1

Reviewer 1 Report

Comments and Suggestions for Authors

The presented research explores the influence of cantilevers on the embodied carbon of structural systems in buildings. The impact is calculated by comparing two building models, the existing structure of the Melbourne School of Design (MSD) building, and its modification by introducing two supporting columns beneath the previously unbalanced cantilevered structure. The investigation revealed that the analyzed cantilever contributed to 2.26% of the total embodied carbon within the whole structural system and, in the case of the segment of the structural system affected by that cantilever, over 10%. Additionally, research showed the significant impact of differing LCI methodologies (particularly between the EPiC database and EPDs) on assessing architectural decisions. The conclusions contained in the article call for a discussion on the architecture evaluation criteria, particularly in the case of buildings with “pedagogical intentions,” and the need to increase the importance of environmental footprint assessment of spectacular construction solutions.

The article is well structured. The cited references are relevant and cover recent publications. The research design is appropriate, with research methods clearly described. The results are clearly presented. The conclusions are consistent with the evidence and arguments presented.

Minor issues should be corrected before the publication of the article:

·      For the consistency of the presented results, in line 613, there should be 2.26% as in line 743. 

·      In Section 5.2, a table or figure comparing the Base Case Model and Supported Cantilever Model would be beneficial for a more straightforward examination of the contributing factors to the increased embodied carbon. In that regard, visualization of the embodied carbon changes per story, per material, and per structural element type, both for the whole building and for the cantilever-influenced structure, would help. Currently, only Table 3 presents the most significant changes without the possibility of close examination.

·      The finite element models and supporting data developed for this study cannot be accessed—the provided URL (https://figshare.com/s/c6f2f28682c7542633e2) links to the page where files have not been added.

Reviewer 2 Report

Comments and Suggestions for Authors

I am personally pleased to see the research on the integration of architecture and structural engineering. Based on the design goal of green building, the research is systematically studies the common problem in the current stage of structural design, and the manuscript logically expresses the specific impact of structural type differences on embodied carbon.

Suggest:  The manuscript has too many sections, please condense the content and merge the first two sections.

Comments on the Quality of English Language

The English expression is fine, and the description is clear.

Author Response

We are grateful for your critical assessment of the manuscript’s structure and your suggestion to condense the content by merging the first two sections. After careful consideration, we have decided to retain the current structure of the paper. We believe that each section serves a distinct and vital purpose within the narrative and logical flow of the paper.

The division of the introduction and literature review into separate sections (Sections 1 and 2) adheres to conventional academic norms, allowing us to clearly establish the study's aims and scope before delving into the pertinent literature. We feel that combining these sections would create an unwieldy and less reader-friendly opening section, potentially obscuring the clarity and focus provided by the current structure.

Thus, we kindly propose to maintain the current format of the paper, which we believe best serves the comprehensive presentation of our research and supports the readers' understanding.

Thank you for your understanding and for your constructive feedback.